# Autoclaved Diet with Inactivated Spores of *Bacillus* spp. Decreased Reproductive Performance of *Muc2^−/−^* and *Muc2^+/−^* Mice

**DOI:** 10.3390/ani12182399

**Published:** 2022-09-13

**Authors:** Maryana V. Morozova, Galina V. Kalmykova, Nadezhda I. Akulova, Yuriy V. Ites, Valentina I. Korkina, Ekaterina A. Litvinova

**Affiliations:** 1Scientific-Research Institute of Neurosciences and Medicine, St. Timakova, 4, 630117 Novosibirsk, Russia; 2Physical Engineering Faculty, Novosibirsk State Technical University, 630073 Novosibirsk, Russia; 3Siberian Federal Scientific Center of Agrobiotechnology RAS, St. Central, 1, 630501 Krasnoobsk, Russia

**Keywords:** autoclaved diet, *Bacillus* spp., mice, reproductive performance, model of inflammatory bowel disease, *Muc2**^−/−^* mice

## Abstract

**Simple Summary:**

Within barrier facilities for the housing of laboratory animals, the sterilization of feed, bedding, and cages is used to reduce contact with bacteria. However, in nature, animals come into contact with a lot of bacteria. We investigated the effect of an autoclaved diet on the reproductive performance of *Muc2^−/−^* mice. *Muc2^−/−^* mice develop intestinal barrier defects and are sensitive to changes of the gut microbiota. We have shown that the autoclaved diet negatively affects the reproductive performance of *Muc2^−/−^* females and their healthy *Muc2^+/−^* siblings. Thus, the autoclaved diet led to earlier rectal prolapse of *Muc2^−/−^* females combined with intestinal inflammation, compared to mice fed with the non-autoclaved diet. We hypothesize that this effect is due to the reduction of the diet nutritional value and inactivation of *Bacillus* spp. spores in the autoclaved diet.

**Abstract:**

Within barrier facilities, autoclaved diet and bedding are used for husbandry of laboratory rodents. *Bacillus* spp. are ubiquitous in nature and some of them are known as probiotics. Inactivation of the *Bacillus* spores and reduction of the diet nutritional value due to autoclavation could be especially critical for immunodeficient mice. We studied the effect of the autoclaved and non-autoclaved diets on the reproductive performance and the age of prolapse manifestation in *Muc2^−/−^* mice with impaired gut barrier function and, therefore, sensitive to change of microbiota. We found that the non-autoclaved diet led to enhancement of the fertility index of *Muc2^−/−^* and *Muc2^+/−^* female mice. The non-autoclaved diet affected the prolapse of *Muc2^−/−^* mice that occurred later in comparison with females eating the autoclaved diet. We showed that *Bacillus* spp. was present in the non-autoclaved diet and feces of mice on the non-autoclaved diet. Bacterial strains of the non-autoclaved diet and feces belonged to *B. amyloliquefaciens*, *B. thuringiensis*, *B. subtilis*, *Lysinibacillus macrolides*, *B. cereus*, and other representatives of *Bacillus* spp. Moreover, autoclavation of the diet affected on the percent of the blood and spleen immune cells, the bacterial composition of the intestine, and increased the level of methionine in the thigh muscle of mice. Enhanced reproductive performance and delayed prolapse manifestation in *Muc2^−/−^* mice could be due to improved digestion, as *Bacillus* spp. from diet and feces had enzymatic activity.

## 1. Introduction

Health and wellness of laboratory animals are indispensable in modern animal research planning. Enrichment of the environment leads to an increase in the quality of life of laboratory animals. Standardization of enrichment will ensure stable and reproducible scientific results. In research experiments, the use of the 3R’s (replacement, refinement, and reduction of animal number) rules has recently begun [1]. Principles of laboratory animals’ housing are being changed; now, preference is given to barrier facilities with a division of traffic flow, which is comfortable both for animals and personnel [2]. Great attention is paid to the control of microbiological contamination, including regular health monitoring, and the sterilization of diet and individual equipment [3]. This approach reduces the infection risk of animals [4]. In addition, the stability of the microbiota of laboratory animals, such as specific pathogen free (SPF) status, may be important for the quality of research and reproducibility of results [5]. However, a sterile diet and microenvironment are not natural for animals. Natural environments contain not only pathogenic, but also potentially probiotic bacteria.

The widespread bacteria are representatives of the genus *Bacillus*. *Bacillus* spp. were isolated from the soil and their spores spread by wind or mechanically with soil particles. It is well known that *Bacillus* spp. are endophytes that reside in plant tissue without causing any harmful effects [6,7,8]. Thus, the natural rodents’ feed is enriched with *Bacillus* spp. bacteria. *Bacillus* spp. are allochthonous microorganisms that enter the gastrointestinal tract of mammals when accidentally eaten [8]. These bacteria do not colonize but move through it and have a probiotic effect on the host organism. It is known that some of *Bacillus* spp., such as *Bacillus subtilis*, *Bacillus coagulans*, and *Bacillus licheniformis* improve digestion and assimilation of feed, growth, and reproductive rates of animals [9,10,11,12]. Some bacteria of *Bacillus cereus* produce toxins and cause poisoning. However, other members of this group have a positive effect on health and are included in commercial probiotic preparations [13,14,15].

Since *Bacillus* spp. forms spores, which are resistant to storage and processing of feed, as well as to the digestive tract conditions [10,16], it can be used as a probiotic [16,17,18]. *Bacillus* spp. demonstrates antibacterial and antimycotic activity and could change the host intestinal microbial composition [9,11,19,20,21]. Obviously, the absence of bacteria that act as probiotics in the diet is critical for transgenic and immunodeficient mice. 

Thus, *Muc2^−/^*^−^ mice, a model of colitis and colorectal cancer, are sensitive to changes in intestinal microbiota [22,23]. *Muc2^−/−^* mice lacking *Mucin-2* (*Muc 2*) demonstrated intestinal inflammation, an increase in gut permeability, diarrhea, and prolapse [23,24,25,26]. Intestinal microbiota of *Muc2^−/^*^−^ mice differed from the microbiota of control mice [27].

In the present study, we showed how the autoclaved and non-autoclaved diets, that harbored *Bacillus* spp., affected the presence of *Bacillus* spp. in feces, on bacterial composition of feces, and on reproductive performance, immunity, blood parameters, and muscle amino acid composition in *Muc2^−/−^* mice with intestinal barrier defects, and in heterozygote siblings with a healthy phenotype.

## 2. Materials and Methods

### 2.1. Animals

The experiments were performed in the Scientific Research Institute of Neurosciences and Medicine (SRINM). Two–three-month-old *Muc2^−/−^* Nskrc and *Muc2^+/−^* Nskrc mice were used. To obtain *Muc2^+/−^* and *Muc2^−/−^* mice, *Muc2^+/−^* females were bred with *Muc2^+/−^* males. Reproductive performance of *Muc2^−/−^* female mice was assessed until the registration of prolapse. The humane endpoint for mice with prolapse was used. Animals were housed in single-sex groups of 2–4 animals or families of two females and one male in individually ventilated cages (Optimice, Animal Care Systems, Centennial, CO, USA). The light/dark photoperiod was 12 h/12 h (light off at 12:00 h), the temperature was 22–24 °C, and the humidity was 30–60%. For cytometry analysis, 4–5 *Muc2^−/−^* mice were used. To determine the intestinal bacterial composition on the autoclaved diet and for 2 weeks on the non-autoclaved one, three and four females, respectively, were used. The switch from the autoclaved to non-autoclaved diet was performed at the age of 6 weeks. To study the effect of long-term feeding with the non-autoclaved diet, five adult females, which had been fed since birth, were used. To study the effect of long-term feeding with the autoclaved diet, three adult females, which had been fed since birth, were used. To analyze the intestinal bacterial composition mice from same breeding family were taken. As it is known that colonization of the intestine by microorganisms starts since birth and during several days after birth [28], it could be assumed that the initial intestinal microbiota was similar in mice from the same breeding family.

The diet (R-22, BioPro, Novosibirsk, Russia) and water (bottled) were provided ad libitum. Feces from *Muc2^−/−^* and *Muc2^+/−^* mice were collected directly from mice after 12–13 weeks of feed of the autoclaved or non-autoclaved diets, and in 10, 20, 30 days after the non-autoclaved diet changed to the autoclaved, or in 2 weeks after changing the autoclaved to the non-autoclaved diet. The mice that were saved in the experiment were used for other experiments. All mice were saved except 10 heterozygotes mice (*Muc2^+/−^*), whose muscle composition of amino acids and blood biochemistry were assayed, and 10 *Muc2^−/−^* mice that were used for immune cells.

All procedures were conducted in accordance with the European Convention for the protection of vertebrate animals used for experimental and other scientific purposes. All procedures were approved by the bioethical committee of Scientific-Research Institute of Neurosciences and Medicine, protocol # 3 (03/11/2021). All animals were specific-pathogen free, which was tested quarterly according to Federation of European laboratory animal science associations (FELASA) recommendations [29].

Autoclavation of the diet, bedding birch sawdust, cages and bottles were carried out at autoclave GE 91415 AR-2 (Getinge Sterilization AB, Gettinge, Gothenburg, Sweden), under pressure of 2 atm. and temperature of 121 °C, for 30 min. The cages and bottles for the group “non-autoclaved diet” were only washed with antiseptic Anavidin (Anavidin, Irkutsk, Russia); as for the “autoclaved diet” group but were not autoclaved. The diet and sawdust for the group “non-autoclaved diet” was untreated. Mice of different groups were kept in the same vivarium, but in different rooms with barrier that prevent the transfer of microorganism between cages.

### 2.2. Reproductive Performance

Reproductive performance was calculated for colonies of female mice that were fed on the autoclaved (*Muc2^+/−^ n* = 74, *Muc2^−/−^ n* = 34) and non-autoclaved diet (*Muc2^+/−^ n* = 21, *Muc2^−/−^ n* = 10). Reproductive performance was calculated as the number of all offspring born in the reproductive period divided by all females in breeding and divided by the number of weeks in the selected reproductive period. The reproductive performance was assayed for 8 weeks—the period between the two following pregnancies of one female. This index reflects the reproductive ability of a female [30].

### 2.3. Analysis of CFU Count

From each sample of the diet, an aliquot (10 g) was homogenized with 90 mL of sterile saline solution (PBS). The aliquots of the mouse feces (100 mg) were homogenized with 0.9 mL of PBS. Subsequently, a decimal dilution series prepared in PBS and 500 μL of the appropriate dilutions were plated on Dextrose Casein-peptone agar (Merck, Darmstadt, Germany). Removal of vegetative forms of bacteria was carried out by heating the homogenate for 15 min at 85 °C. The plates were incubated aerobically at 36 °C for 48 h.

### 2.4. Microscopy

Spore formation in the bacterial cultures was confirmed by microscopy. A drop of saline solution was applied to a glass slide, and a bacterial culture was introduced into it with a bacteriological loop. The smears were dried and fixed. Then, methylene blue was applied to the sample. After 10 min, the smears were washed with water to remove the rest of methylene blue.

Crystal formation was studied on fixed preparations stained with 5.0% aqueous solution of carbolic eosin. The carbolic eosin was spread on the surface of the smear and heated on the flame until the point of an intense discharge of fumes. After 2 min, the smear were washed in tap water and drained.

Stained glass slides were examined under an oil immersion microscope (Olympus Corp., Tokyo, Japan) at 100× resolution.

### 2.5. Determination of the Enzymatic Activity of Bacterial Isolates

Amylolytic activity was tested by hydrolysis of starch added to the agar medium by bacterial cells. After incubation, the agar was filled with Lugol’s solution. A positive reaction was judged by the appearance of a colorless area around the growth zone of the colony.

Proteolytic properties were determined by the zones of hydrolysis (clear zones) around the colonies on skim milk media.

Lipase activity was analyzed by the appearance of a cloudy halo around colonies grown on nutrient agar with the addition of 0.01% CaCl_2_ and 1% Tween 80.

### 2.6. qPCR Analysis

DNA was isolated from the mice feces using a QIAamp DNA Stool Mini Kit (Qiagen, Hilden, Germany) according to the manufacturer’s recommendations. The amount of bacterial DNA was determined by the 16S rRNA gene region by real-time PCR using BioMaster HS-qPCR SYBR Blue (BioLabMix, Novosibirsk, Russia), 5 μL DNA from feces, and 300 nM specific primer (Appendix A). The negative control was deionized water (Milli-Q type I, Merck Millipore, Darmstadt, Germany). The amplification reaction was run in a CFX96 real-time PCR instrument (Bio-Rad, Hercules, CA, USA). DNA was denatured for 5 min at 95 °C and then amplified in 45 denaturation cycles at 95 °C for 15 s, then primer annealing and DNA synthesis at 62 °C for 50 s. Bacterial DNA was normalized to the 16S rRNA gene as follows: log10[δCt × (16S rRNA of each bacterial group/total 16S rRNA)].

### 2.7. Sanger Sequencing of the 16S rRNA Gene

To obtain cultures, bacteria were purified by repeated streaking and single colony culture at 37 °C for 48 h. Bacterial suspensions prepared from the individual colonies were boiled and lysates were used as a template to amplify the 16S rRNA gene. PCR was performed using the primers 16S_27F (AGAGTTTGATCMTGGCTCAG) and 16S_1492R (GGTTACCTTGTTACGACTT). The sequencing of the resulting PCR fragments was performed using the same primers and a Terminator v.3.1 BigDyeTM kit (Applied Biosystems, Waltham, MA, USA) according to the manufacturer’s protocols. Capillary electrophoresis was performed on ABI 3130xl Genetic Analyzer (Applied Biosystems, Waltham, MA, USA) [31].

### 2.8. Cytometry of Blood and Spleen Cells 

Blood samples were collected from the retroorbital sinus without the use of anesthesia, which can affect the immune cell count by release glucocorticoids. Ophthalmic anesthetic (0.5% proparacaine hydrochloride ophthalmic solution, Alcon Laboratories, Alcon-Couvreur, Brussels, Belgium) was used for treatment of mice eyes. Then, the mice were euthanized by CO_2_. Spleens were placed and stored in plastic dishes (diameter, 50 mm) with cold PBS. For flow cytometry, red blood cells were lysed by ammonium chloride (0.15 M)–potassium carbonate (1 mM) buffer and homogenized; the homogenate was filtered through 70-μm cell strainers (catalog no. CLS431751, Corning, New York, NY, USA). The spleen was lysed, and splenocytes were washed twice. Blood and spleen cells were stained with PE–anti CD3ε (hamster, clone 145-2C11), FITC–anti CD4 (rat IgG2b κ, clone GK1.5), PE/Cyanin7–anti CD8a (rat IgG2a κ, clone 53-6.7), PE–anti CD3ε (Armenian hamster IgG, clone 145 to 2C11), and FITC–anti CD19 (rat IgG2a κ, clone 6D5) antimouse antibodies (BioLegend, Dedham, MA, USA) for 120 min at 4 °C in the dark and then analyzed by using a BD FACSCanto II Flow Cytometer. For analysis of blood cells, singlets were gated on lymphocytes, and CD3+ and CD19+ lymphocytes were gated on lymphocyte singlets or CD4+ and CD8+ cells were gated on CD3+ lymphocyte singlets; 25,000 singlets were counted for each sample. The gating strategy for the analysis of CD3+ and CD19+ splenocytes was the same as for blood samples; 60,000 splenocytes singlets were counted for each sample. The populations of CD3+, CD19+, CD3+CD4+, and CD3+CD8+ cells were determined as a percentage.

### 2.9. Biochemical Analysis of Blood Serum

Blood was collected from the retroorbital sinus of *Muc2^+/−^* females as described previously. Serum creatinine, total protein, low density lipoproteins, and glucose were determined using commercial kits (Applied Biosystems, Waltham, MA, USA) according to the manufacturer’s protocols.

### 2.10. Analysis of the Amino Acid Composition of the Thigh Muscle

Muscle samples were trimmed of visible fat and external connective tissue. Two samples (100 mg) were taken from each mouse to determine amino acids according to the following schemes—with and without derivatization. Capillary electrophoresis was performed on KAPEL^®^-105M (Lumex, Irkutsk, Russia) with a UV detector. Samples were analyzed using cartridge with a quartz capillaries 75 cm long and 50 µm in inner diameter (Lumex, Irkutsk, Russia). The external temperature of the capillary column was set at 30 °C. The capillary electrophoretic separations were performed in electrolyte (phosphoric buffer and cyclodextrin, pH 7.7). Determination of arginine, lysine, tyrosine, phenylalanine, histidine, sums of leucine and isoleucine, methionine, valine, proline, threonine, serine, alanine, and glycine were carried out in phenylisothiocyanate (PITC)—derivatives of amino acids. To obtain PITC, the muscle samples (100 mg) were diluted with 50% HCL to 10 mL (SoyuzKhimProm, Novosibirsk, Russia). Hydrolysis was carried out at a temperature of 110 °C for 14–16 h. Filtered hydrolysates (0.05 mL) were then derivatized with 0.15 mL of 0.1 M Na_2_CO_3_ (SoyuzKhimProm, Novosibirsk, Russia) in 0.3 mL of isothiocyanate solution in isopropanol (0.4 mL isothiocyanate and 21 mL isopropanol) (SoyuzKhimProm, Novosibirsk, Russia). After 35 min, the derivatives were dried with warm air. Dried precipitates were diluted with distilled water (0.5 mL). A total of 500 μL of solution was added into the autosampler. Detection was carried out in the UV region of the spectrum at a wavelength of 254 nm. Samples were injected applying a pressure of 30 mbar, Voltage: +25 kV. The mass fraction of tryptophan was found in the muscle sample’s liquid fraction directly without obtaining PITC-derivatives. To obtain a liquid fraction, the 5 mL of barium hydroxide octagonal crystalline hydrate solution (SoyuzKhimProm, Novosibirsk, Russia) was added to muscle (100 mg). The electrolyte was made of sodium tetraborate (10 mL, 0.05 M, pH 9.2) and 15 mL of distilled water. Detection was carried at a wavelength of 219 nm. Samples were injected applying a pressure of 30 mbar, Voltage: +25 kV. A commercial amino acid standard mixture (catalog no. LAA21-1KT, Sigma Aldrich, Darmstadt, Germany), was used as an external standard. Post-run analysis of the data was performed with specialized software “Elforan” (Lumex, Irkutsk, Russia). After each run, the capillary column was washed with fresh electrolyte for 3 min.

### 2.11. Analysis of the Diet Mineral and Chemical Composition 

Phosphorus contents of autoclaved and non-autoclaved samples were determined by atomic absorption spectrometry (Shimadzu AA-700, Kyoto, Japan) after digestion in nitric acid, percholoric acid, and hydrochloric acid at 440 nm. Calcium was measured by titration with Trilon-B solution. Total protein content (N × 6.25) was determined using an automatic Kjeldahl system (230-Hjeltec Analyzer; Foss Tecator, Hoganas, Sweden), total lipid was determined with an automatic Soxhlet system (2050-FOSS; Sweden) by chloroform extraction, moisture was determined by drying at 105 °C for 24 h in an oven (D-63450; Heraeus, Hanau, Germany), and ash was determined by incinerating in a muffle furnace (Isuzu, Tokyo, Japan) at 550 °C for 6 h. Fiber was determined by the Weende method on a Semi-Automatic Fiber Analyzer (Velp Scientifica, Usmat, Italy). Starch and sugar were determined by the Anthrone method using an Easy Brix refractometer (Mettler Toledo, Greifensee, Switzerland).

### 2.12. Statistical Analysis 

The data are presented as means ± SEMs for quantitative values and as percentages for qualitative ones. Distribution was determined using descriptive statistics in the Statistica 10.0 program using the Kolmogorov–Smirnov test. The data with a non-normal distribution were assessed by Mann–Whitney *U* test for independent groups. For categorical variables, the differences between experimental groups were analyzed by Fisher’s exact test. The value of *p* < 0.05 was considered significant.

## 3. Results

### 3.1. Reproductive Performance and Age of Prolapse of Muc2^−/−^ and Muc2^+/−^ Female Mice

Husbandry of the laboratory animals with a sterilized diet and bedding is not natural for them. We compared the reproductive and physiological parameters of mice that were housed either in a sterilized cage with the autoclaved diet and bedding or close to natural, in which the diet and bedding were not autoclaved. We compared the physiological parameters of two strains: heterozygous *Muc2^+/−^* females, which have never developed intestinal pathology, and knockout *Muc2*^−/−^ females, which have prolapse, reduced weight, and blood in the feces. To compare the reproductive performance of mice housing under different conditions, the fertility index was calculated using the formula: the number of all born pups divided by all breeding females during 8 weeks.

The reproductive index was significantly higher for heterozygous and knockout mice, which were fed on the non-autoclaved diet (*p* < 0.001, Fisher’s exact test, Figure 1A). However, the reproductive performance of *Muc2*^+/−^ females was significantly higher compared to *Muc2*^−/−^ females fed on both the autoclaved (*p* < 0.001, Fisher’s exact test, Figure 1A) and non-autoclaved diets (*p* < 0.001, Fisher’s exact test, Figure 1A). Thus, reproductive performance was higher in mice of both genotypes fed on the non-autoclaved diet. The development of intestinal inflammation, which eventually leads to prolapse, characterizes *Muc2*^−/−^ mice. Prolapse develops in reproductively active mice, could reduce the likelihood of pregnancy, and also affects the nursing of offspring by mothers.

Prolapse developing in *Muc**2^−/−^* mature female mice reduces the chance of giving birth to offspring. We compared the age of prolapse onset in *Muc**2^−/−^* females fed on the autoclaved or non-autoclaved diets. Female mice fed on the autoclaved diet showed signs of prolapse in three months age, in comparison to females fed on the non-autoclaved diet that developed prolapse only at six months (Z = 3.71, *p* < 0.001, Figure 1B). Mice fed on the autoclaved diet until four months age developed prolapse and could not give birth to offspring. There were no *Muc**2^−/−^* females fed on the non-autoclaved diet until the same age with prolapse manifestation (*p* < 0.05, Fisher’s exact test, Figure 1C). Thus, feeding on the autoclaved diet reduced the number of mature *Muc**2^−/−^* female mice with prolapse and this affected their reproductive performance.

### 3.2. Count of CFU Bacillus spp. Spores and Morphological Characterization of Bacterial Colonies in the Diet and Feces of Mice

A diet for laboratory mice is made from raw plants that could include bacteria, for example, *Bacillus* spp. Autoclaving inactivates not only vegetative bacterial forms, but also spores. We hypothesized that bacteria of the genus *Bacillus* may reduce inflammation and enhance the reproductive performance of mature females fed on the non-autoclaved diet. We tested the autoclaved and non-autoclaved diets for the presence of *Bacillus* spp. spores. To inactivate vegetative forms of the bacteria, we heat the suspension of the diet for 15 min at 85 °C. Then, inactivated suspension was cultured on the plates with Dextrose Casein-peptone agar, the preferred culture medium for *Bacillus* spp. As a result, we found several colonies related to *Bacillus* spp. by morphological characterization (a more detailed study of the diet was published earlier [32]). Analysis of bacterial colonies of *Bacillus* spp. from the non-autoclaved diet has shown 4.43 ± 0.41 × 10^3^ CFU per gram (Figure 2A), which, according to morphological characterization, can be attributed to the genus *Bacillus* [33]. Microscopic examination of bacterial colonies showed that all bacteria in the colonies are oblong and form oblong spores (a more detailed study of the diet was published earlier [32]). We did not find bacterial growth in the autoclaved diet. Thus, only the non-autoclaved diet contained bacterial spores that formed the colonies by morphological characterization of *Bacillus* spp., and autoclaving inactivated them. Analysis of the bedding before and after autoclaving did not show growth of the *Bacillus* spp.

Consumption of spores of *Bacillus* spp. with a non-autoclaved diet could lead to bacterial entry into the gut microbiota. This means that spores of *Bacillus* spp. could be present in the feces of the mice. We supposed that autoclaved diet would not promote *Bacillus* spp. in feces. Indeed, there were no spores of *Bacillus* spp. in the feces of the mice of both genotypes fed on the autoclaved diet. Analysis of bacterial colonies of fecal samples of *Muc2^−/−^* mice fed on the non-autoclaved diet showed 3.9 ± 0.5 × 10^4^ CFU/g (Figure 2A). It is known that that intestinal microbiota of the *Muc2^−/−^* mice, which lack the mucin 2, differs from microbiota of wild-type mice [27]. Feces of the *Muc2^+/−^* mice had two times more spores—9.75 ± 0.2 × 10^4^ CFU/ g, but there was not a significant difference in the comparison with *Muc2^−/−^* mice (*p* > 0.05, Fisher’s exact test, Figure 2A).

### 3.3. Speed of Bacillus spp. Elimination from the Feces of Mice Switched to the Autoclaved Diet

*Bacillus* spp. are allochthonous bacteria; therefore, when their source disappears, count of bacteria in the intestine should decrease until complete cleansing. To understand how quickly *Bacillus* spp. would eliminate from the digestive tract of mice, we examined feces for presence of *Bacillus* spp. after 10, 20, and 30 days of feeding on the autoclaved diet.

After 10 days, the number of spores in the feces of *Muc2^−/−^* mice decreased more than 100 times—2 *±* 0.05 × 10^2^ CFU/g (*p* < 0.001, Fisher’s exact test), after 20 days decreased 10 times—4 *±* 0.02 CFU/g (*p* < 0.001, Fisher’s exact test), there were no spores in feces after 30 days. The feces of *Muc2^+/−^* mice were analyzed for the present of spores in the same time points. After 10 days, the number of spores in the feces was 3 *±* 0.05 × 10^2^ CFU/g, after 20 days—9 *±* 0.02 CFU/g, and after 30 days there was no growth of colonies (Figure 2B).

After two weeks of the non-autoclaved diet consumption in *Muc2^+/−^* mice that previously were fed on the autoclaved diet, spores of *Bacillus* spp. appeared in feces and reached 2.03 ± 0.04 × 10^4^ CFU/g. The number of bacterial spores from the feces of *Muc2^+/−^* mice fed on the non-autoclaved diet since birth was the same as from feces of mice that two weeks fed on the non-autoclaved diet. Indeed, *Bacillus* spp. are not found in the feces of mice if their spores are not present in the diet.

### 3.4. Identification of Bacterial Strains from the Diet and Feces

For more accurate identification of bacteria, we analyzed the sequences of the 16S rRNA gene of bacteria from four strains from the diets and five strains from the feces. All bacterial strains were confirmed as belonging to the genus *Bacillus*. The first strain from the diet with 100% identity refers to the *Bacillus subtilis* strain qx-4 (Accession number MW221326) and *Endophytic bacterium* MD3 (Accession number HM160161), the second with 99.93% identity refers to the *Bacillus thuringiensis* strain GZDF1 (Accession number MT358632) and 100% identity to *Bacterium* strain MIS_YL_J55 (Accession number MW037810), the third with 100% identity to the *Lysinibacillus* sp. FWQSR5 (Accession number MN647595) and *Lysinibacillus macroides* strain Z010 (Accession number MG266471), the fourth with 100% identity to the *Bacillus cereus* strain S43 (Accession number KP279290), with 99.68% identity to *Bacillus thuringiensis* strain BT62 chromosome, complete genome (Accession number CP044978 QAPB01000000 QAPB01000001-QAPB01000147), and with 100% identity to the *Bacillus paramycoides* strain HBU72519 (Accession number MW365218). The strains from the feces belonged to the same genus as the strains from the diet—*Bacillus*. The first strain from feces is 99.86% identical to the *Bacillus* amyloliquefaciens strain QT-162 (Accession number MT081100) and *Bacillus* velezensis strain 2645 (Accession number MT611666.1), the second one is 100% identical to the *Bacillus* thuringiensis strain GZDF1 (Accession number KP137560) and *Bacillus* sp. (in: Bacteria) strain 201705CJKOP-59 (Accession number MG309372), the third isolate is 99.93% identical to the *Bacillus subtilis* subsp. stercoris strain EGI137 (Accession number MN704441) and *Bacillus subtilis* strain a22 (Accession number MK726116), the fourth strain is 98.53% identical to the *Bacillus* cereus strain YB1806 (Accession number MH633904), to the *Bacillus* parathracis (Accession number MT422123), *Bacillus* thuringiensis strain NO.8 (Accession number MN509082), the fifth with 99% identical to the *Bacillus* cereus strain D21 (Accession number KC441762), to the *Bacillus* parathracis, (Accession number MT422123), and to the *Bacillus* cereus strain YB1806 (Accession number MH633904).

The bacteria *Bacillus cereus* and *Bacillus thuringiensis* are genetically very close and poorly differentiated; however, *Bacillus thuringiensis* forms crystals that are visible under a light microscope on a fixed and stained carbolic eosin preparation. Microscopy of bacteria from feces showed that all strains except one of the feces produced para-sporal crystal inclusions and could, therefore, be identified as *Bacillus thuringiensis*. Thus, we attributed to *Bacillus cereus* only one strain isolated from feces. We concluded that bacterial strains from the diet and feces belonged to the class *Bacilli*, order *Bacillales*, family *Bacillaceae.* Housing mice on a non-autoclaved diet has a positive effect on the reproductive health of *Muc2^−/−^* and *Muc2*^+/−^ mice and the state of the intestines of *Muc2^−/−^* mice, characterized by the later prolapse. We believe that the positive effect on health may be associated with the probiotic effect of *Bacillus* spp. The mechanism mediating the beneficial effects of *Bacillus* spp. on the host organism may be associated with a change of intestinal microbiota or with a change of the immune profile [34]. To test the hypothesis about the change of microbiota, we analyzed the representatives of the commensal bacteria of the *Muc2^+/−^* mice, which were fed on the autoclaved or non-autoclaved diets. 

### 3.5. Fecal Bacteria Composition of Mice Fed on Autoclaved and Non-Autoclaved Diets

We did not analyze fecal bacterial composition of *Muc2^−/−^* mice because the Mucin2 absence could change the bacterial composition. We analyzed bacteria of feces in three groups of mice: the first group was fed on the autoclaved diet since birth, the second group was fed on the non-autoclaved diet since birth, and the third group of mature mice that was fed on the non-autoclaved diet for 2 weeks. Two weeks of feeding on the non-autoclaved diet caused a decrease in the amount of the *E. coli* 16S rRNA gene (Z = 2.12, *p* < 0.05, Figure 2C). The amount of 16S rRNA of *A.muciniphila*, *E.faecalis*, *Lactobacillus* spp., *Bacteroides* spp., and *Staphylococcus* spp. in feces did not change after 2 weeks of feeding on the non-autoclaved diet that contained spores of *Baccilus* spp. (*p* > 0.05) (Appendix A).

Feeding of the non-autoclaved diet since birth led to a significant decrease in the 16S rRNA of *Bacteroides* spp. (Z = 2.23, *p* < 0.05) and *Lactobacillus* spp. (Z = 2.23, *p* < 0.05) in feces compared to mice fed on the autoclaved diet free from *Baccilus* spp. There were no significant differences in the amount of the 16S rRNA gene of *A. muciniphila*, *E. faecalis*, *E. coli* and *Staphylococcus* spp. in feces of mice fed on the non-autoclaved diet (*p* > 0.05) (Appendix A). Therefore, the non-autoclaved diet rich of *Bacillus* spp. altered the bacterial community.

### 3.6. Immune Cell of Mice Fed on the Autoclaved and Non-Autoclaved Diet

To assess the effect of the autoclaved diet on the blood immune cells there were determined percent of B cells (CD19+), T helper (CD3+CD4+) and T killer (CD3+CD8+) cells (Figure 3A). *Muc2^−/−^* mice fed on the non-autoclaved diet increased the percentages of CD3+CD8+ cells (Z = 1.22, *p* < 0.05) (Appendix A, Figure 3). The non-autoclaved diet enhanced in spleen the percentage of CD3+CD4+ (Z = 1.19, *p* < 0.05) and increased CD19+ cells (Z = −1.98, *p* < 0.05). (Figure 3B). There was no significant effect on the percentage of CD3+ and CD3+CD8+ cells in spleen of *Muc2^−/−^* mice fed on the non-autoclaved diet (*p* > 0.05) (Appendix A).

Thus, the consumption of the non-autoclaved diet containing *Bacillus* spp. spores causes a decrease of *Bacteroides* spp. and *Lactobacillus* spp. in mice feces, as well as a decrease of the percent of T helper and B cells in the blood, which are involved in the humoral immune response. The bacteria *Bacillus* spp. are known for their ability to synthesize digestive enzymes and improve host digestion [35,36,37,38]. Therefore, bacterial strains from the diet and feces were tested for amylase, proteolytic, and lipolytic activities.

### 3.7. Enzymatic Activity of Bacteria from the Diet and Feces—Blood Biochemical Analysis of Muc2^+/−^ Mice Fed on Autoclaved and Non-Autoclaved Diets

All bacterial strains from the feces and diet had enzymatic activity and could take part in the digestion (Table 1, Appendix A). 

We assumed that such enzymatic activity has a positive effect on the absorption of nutrients and also affects the metabolism of mice. Therefore, we assayed the level of the following metabolites in the blood: glucose, creatinine, total protein, low-density lipoproteins. The data are presented in Table 2. We showed that in *Muc2^+/−^* females fed on the autoclaved diet stable levels of creatinine, glucose, proteins, and low-density lipoproteins were maintained. Thus, despite the detected enzymatic activity of *Bacillus* spp. from the diet and feces, there were no significant differences in the levels of the blood metabolites.

### 3.8. Amino Acid Composition of the Thigh Muscle of Muc2^+/−^ Mice

Amino acid composition of the thigh muscle showed increased levels of methionine (%) in *Muc2^+/−^* mice fed on the autoclaved diet (Z = −2.40, *p* < 0.05) (Figure 3C). There were no significant differences in the levels of arginine, lysine, tyrosine, phenylalanine, histidine, leucine-isoleucine, valine, proline, threonine, serine, alanine, glycine, and tryptophan (Z = −0.73, Z = −0.73, Z = −1.15, Z = −0.94, Z = 0.31, Z = −0.73, Z = −2.40, Z = 0.00, Z = −0.31, Z = −1.76, Z = −1.37, Z = −0.94, Z = −0.73, and Z = 1.04, *p* > 0.05, respectively, Appendix A). Amino acids in the thigh muscle of mice did not depend on presence or absence of *Bacillus* spp. The exception is methionine, which is elevated in mice on the autoclaved diet.

### 3.9. Analysis of the Mineral and Chemical Composition of the Diet

Analysis of the feed for amino acids did not show the influence of autoclaving on arginine, lysine, tyrosine, phenylalanine, histidine, leucine-isoleucine, valine, proline, threonine, serine, alanine, glycine, glutamine, asparagine, and cystine levels after autoclaving (Z = −1.53, Z = 1.53, Z = 0.65, Z = 1.53, Z = −1.53, Z = 0.65, Z = 1.52, Z = 0.65, Z = 1.52, Z = 1.53, Z = 1.53, Z = 0.6553, Z = 0.65, Z = −1.53, and Z = 0.65, *p* > 0.05, respectively). The amount of methionine and tryptophan was decreased after autoclaving at the trend level (Z = 1.96, Z = −1.96, *p* = 0.05, respectively, Appendix A). 

Feed testing for dry matter (%), moisture (%), crude protein (%), crude fat (%), crude fiber (%), sugar (%), starch (%), calcium (%), and phosphorus (%), also showed no effect from autoclaving (Z = −0.65, Z = −0.65, Z = 0.65, Z = −1.53, Z = 1.53, Z = −0.65, Z = 0.65, and Z = 1.53, *p* > 0.05, respectively). However, the raw ash content was reduced in the feed after autoclaving at trend level (Z = 1.96, *p* = 0.05, Appendix A).

Thus, both in *Muc2^−/−^* and *Muc2^+/−^* mice autoclaved diet significantly reduced reproductive performance and increased pathological defect of the rectum, which was manifested by early prolapse. The presence of *Bacillus* spp. in the non-autoclaved diet was confirmed by microbiological assay. Being an allochthonous, *Bacillus* spp. appeared in the feces of mice that were fed on the non-autoclaved diet for two weeks. In mice fed on the autoclaved diet since birth, the bacterial composition of the intestinal microbiota changes and the amount of *Bacteroides* spp. and *Lactobacillus* spp. increased. The feeding on an autoclaved diet since birth affects the spleen immune cells of mice, increasing T and B cells percentage and declining T helper cells numbers. Sequencing of bacterial strains from the diet and feces confirmed that they all belonged to the *Bacillus* genus. The presence of amylase, protease, and lipase activity of isolated *Bacillus* strains can improve the absorption of nutrients from food, which can affect the metabolism of the body and increase the fertility of females. However, the presence of *Bacillus* in the diet and intestines did not change the following biochemical parameters in the blood and tissues: creatinine, glucose, protein, and low-density lipoproteins, and amino acids in muscles. Only methionine, which reduces autophagy and accelerates the aging process, decreased in the animals fed on the non-autoclaved diet. As autoclaving reduced crude ash, methionine, and tryptophan in the diet at trend level, it could be also critical for reproduction performance and manifested by early prolapse.

## 4. Discussion

The effects of housing conditions (autoclaved and non-autoclaved diets) on the reproductive performance of *Muc2^−/−^* females with defect of intestinal barrier function and physically healthy of *Muc2^+/−^* females were studied. The intestinal microbiota of mammals consists mainly of bacteria belonging to the *Firmicutes* and *Bacteroidetes* (90%), and a lesser to the *Actinobacteria*, *Proteobacteria*, *Fusobacteria*, and *Verrucomicrobia* [39]. These are autochthonous microorganisms, i.e., resident organisms [40]. Some microorganisms of intestine swallowed accidentally along with food, water, or air pass transiently and do not colonize the intestines. However, they play an important role in the structure of the gut microbiota [41,42].

The deviations of composition and diversity of the intestinal microbiota lead to disorders, primarily of the intestine (inflammatory bowel disease, irritable bowel syndrome, colorectal cancer) [43,44,45], type 2 diabetes, obesity [46], and cardiovascular diseases (atherosclerosis, hypertension, heart failure) [47]. Therefore, a limitation of the microbial load on mice in barrier facilities by sterilizing the diet, water, and bedding alters the microbial composition of the intestine and could have a negative effect on laboratory animals.

It had previously been shown that antibiotic treatment of *Muc2^−/−^* mice made to eliminate *Helicobacter* spp. Led to decline of reproductive performance [48]. This is consistent with our results, which showed the decrease of reproductive rates of *Muc2^−/−^* females fed on the autoclaved diet. This decrease is due to the early onset of intestinal prolapse of *Muc2^−/−^* mice. However, decrease of the reproductive index of *Muc2^+/−^* mice fed on the autoclaved diet indicates that there are other reasons. There are many facts that spores of *Bacillus* spp. In the diet have positive effects on reproductive functions [49,50,51]. This allows us to assume that the decrease in reproductive functions in animals of different genotypes on the autoclaved diet is associated with inactivation of *Bacillus* spp. Spores.

Increased intestinal inflammation, which was manifested by early prolapse of *Muc2^−/−^* animals fed on the autoclaved diet, can also be caused by the absence of viable spores of *Bacillus* spp. It was shown that the bacteria *Bacillus* spp. Are effective in the prevention and treatment of intestinal diseases such as diarrhea, colitis, irritable bowel syndrome, and colorectal cancer [52,53,54,55,56,57]. *Bacillus* spores from food can grow in the gastrointestinal tract and again form spores from vegetative cells [58,59]. This may explain why complete elimination from *Bacillus* spores took about 20–30 days. This period significantly exceeded the time of complete passage of food through the digestive tract. In another study, a complete gut clearance from *Bacillus* spores occurred in 3 weeks [60].

*Bacillus* can change the composition of the gut microbiota by releasing bacteriocins, peptide, and lipopeptide antibiotics [16,37,61,62], affecting the adhesion of other microorganisms [16] and changing the pH of the environment [63]. Changes in the gut microbiota fatally lead to abnormalities in the host’s metabolic and immune profile [64,65]. In addition, *Bacillus* can directly affect the host’s immunity. For example, spores of *B. subtilis* play an important role in the development of intestinal lymphoid tissue. Thus, spore of *Bacillus* in gut affected on the diversity of the primary antibody population in rabbits [66], inhibited phospholipase A2, resulting in suppression of pro-inflammatory cytokines and an increase in anti-inflammatory cytokines [56]. Therefore, we investigated the effect of the autoclaved diet and non-autoclaved diet on the commensal microorganisms and the lymphocytes in blood and spleen cells. It is known that microbiota inhabits the intestine in early ontogenesis [28]. To study the effect of the autoclaved diet on the gut microbiota, we used sibling mice fed on the non-autoclaved diet since birth and on the autoclaved diet for 2 weeks. The initial intestinal microbiota was the same in mice of different groups. Two weeks feeding on the non-autoclaved diet decreased only the quantity of *E. coli*. Feeding on the non-autoclaved diet since birth decreased two genera of bacteria: *Bacteroides* spp. and *Lactobacillus* spp. We assumed that continued breeding on the non-autoclaved diet would make changes more pronounced. It was shown [67], that the number of *Bacteroides* and *Lactobacillus* in the fecal bacterial community structure of broiler chickens decreased when *B.amyloliquefaciens* was used. Another study also showed that *B. subtilis* is able to inhibit enterotoxic *E. coli* infection [68]. For instance, the alterations in intestinal microbiota that we discovered are consistent with the findings of other investigators.

The autoclaved diet increased the percent of T helper and B cells in blood and spleen. This may indicate an increase in antibody formation, and may also lead to the development of allergic reactions [69,70]. Changes in the bacterial composition of the intestine and the number of immune cells indicate the response of mice to differences in the diets. These facts should be taken into account when planning experiments, for example, those involving the purchase of animals in a barrier facility and transferring them to a non-autoclaved diet.

It can be assumed that on the diets with or without *Bacillus*, the microbiota and the immune system of mice come to some equilibrium, adjusting to the existing conditions. However, *Bacillus* is also involved in the metabolism due to digestion of nutrients by releasing proteases, lipases, and amylases [35,36,37,38]. Maximum absorption of nutrients leads to sufficient energy balance in cells and its efficient functioning. It was shown that in *Muc2^−/−^* mice fed on the autoclaved diet, the functions of mitochondria in the intestinal epithelium were impaired [25]. Lack of additional digestive aid is difficult to compensate and can be a key negative factor on the health of mice fed on an autoclaved diet. We have not studied mitochondrial function in mice fed on a non-autoclaved diet. Such studies need to be performed in the future, using probiotic strains of *Bacillus*. However, housing on both autoclaved and non-autoclaved diets did not affect the level of major blood metabolites and amino acid composition of muscles, which confirms that the organism is able to normalize the blood and tissue composition. It is interesting to note that only methionine increased in mice fed on the autoclaved diets. It is known that high level of methionine can reduce autophagy and accelerate aging [71,72]. The increase of methionine in mice fed on the autoclaved diets did not depend on changes of the methionine level in the diet, since after autoclaving the level of methionine was even slightly lower in the diets.

*Bacillus* spp. is widely used as probiotic. We assume that the mice diet contained some of these species. Microscopical analysis and identification of bacteria by morphology of colony and is important but not always reliable. Therefore, sequence by Sanger of 16S rRNA gene of bacterial cultures isolated from the diet and feces were used. It was confirmed that all isolated bacterial cultures were *Bacillus* spp., with an identity of 99% or more [73]. Sequence by Sanger showed that the feces and diet contained the bacteria *Bacillus* spp. The microbiota composition of different aliquots of food and feces may differ. In addition, since colonies were manually selected for sequencing by morphotype and microscopy, visually similar colonies might not be included in the analysis. We have shown that the non-autoclaved bedding is free of *Bacillus* spp. bacteria. Our results confirm that *Bacillus* spp. are not residents of the digestive tract but enter it from the external environment. The results show the possible diversity of *Bacillus* spp. in the diet. Some of the bacterial species identified as probiotic are *B. amyloliquefaciens*, *B. velezensis*, *B. subtilis*, and *Lysinibacillus macroides* [16,37,62,74,75]. *B. thuringiensis, B. paramycoides,* and *Bacillus paranthracis* belong to the *Bacillus cereus* group. Among the representatives of this group there are both probiotic and toxic strains [76]. It is known that the toxic amount of *B. cereus* in a diet is 10^5^–10^8^ CFU/g [77]. In the diet, the total number of *Bacillus* spores was at the level of 10^3^–10^4^ CFU/g. Previously, it had been shown that only 3 of 13 strains (about 20%) from the diet exhibited lecithinase activity, a trait of *B. cereus* [32]. Therefore, presence of toxic strains in the diet is not enough to cause toxicity. Since *B. cereus* and *B. thuringiensis* are genetically close, for more accurate differentiation of bacterial strains from feces they were examined for crystal formation. All samples of bacteria from the diet formed para-sporal inclusions and can be classified as *B. thuringiensis*. These crystals contain proteins that are used as insecticides but are considered harmless to mammals [78]. This result is consistent with earlier results regarding 20% *B. cereus* content in the diet. Only one strain from the feces did not form crystals and could be assigned to *B. cereus*. It was found that all representatives of *Bacillus* spp. entered the intestines of these mice with the diet.

## 5. Conclusions

Thus, while a barrier facility that uses an autoclaved diet reduces the number of factors that could affect the reproduction of the results and limits the emergence and spread of infectious diseases among animals, these conditions negatively affect the reproductive performance of mice. This effect is associated with the presence of genus *Bacillus* in a non-autoclaved diet. It is known that autoclaving reduces the nutritional value of feed, in particular, destroys vitamins. However, our results did not reveal significant reductions in amino acids, crude protein, crude fat, etc., in the diet after autoclaving. Only methionine, tryptophan, and raw ash levels were reduced at trend level. It can be assumed that both factors may affect the reduced reproductive function of mice on autoclaved food: a relative decrease in the nutritional value of feed and inactivation of spores of beneficial *Bacillus* strains. The age of onset of prolapse in *Muc2^−/−^* mice was significantly affected by the presence or absence of *Bacillus* spores.

We believe that further research on the addition of *Bacillus* probiotic cultures or their metabolites to an autoclaved laboratory animal diet would be interesting and useful.

## Figures and Tables

**Figure 1 animals-12-02399-f001:**
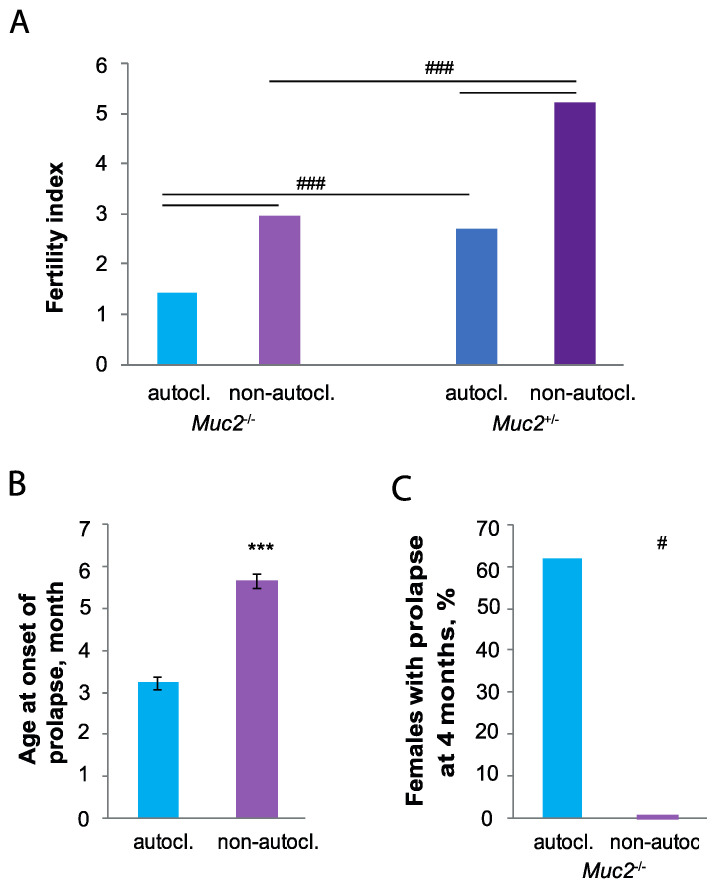
Effect of the autoclaved diet on the reproductive performance and age at onset of prolapse of *Muc2^+/−^* and *Muc**2^−/−^* female mice. (**A**) Fertility index of female mice fed on the autoclaved (*Muc2^+/−^ n* = 74, *Muc2^−/−^ n* = 34) and non-autoclaved diets (*Muc2^+/−^ n* = 21, *Muc2^−/−^ n* = 10). (**B**) Age at onset of prolapse of *Muc**2^−/−^* female mice fed on the autoclaved (*n* = 23) and non-autoclaved diets (*n* = 20). (**C**) Number of *Muc**2^−/−^* female mice with prolapse at 4 months fed on the autoclaved and non-autoclaved diets. # *p* < 0.05, ### *p* < 0.001, Fisher’s exact test, *** *p* < 0.001 Mann–Whitney *U*-test.

**Figure 2 animals-12-02399-f002:**
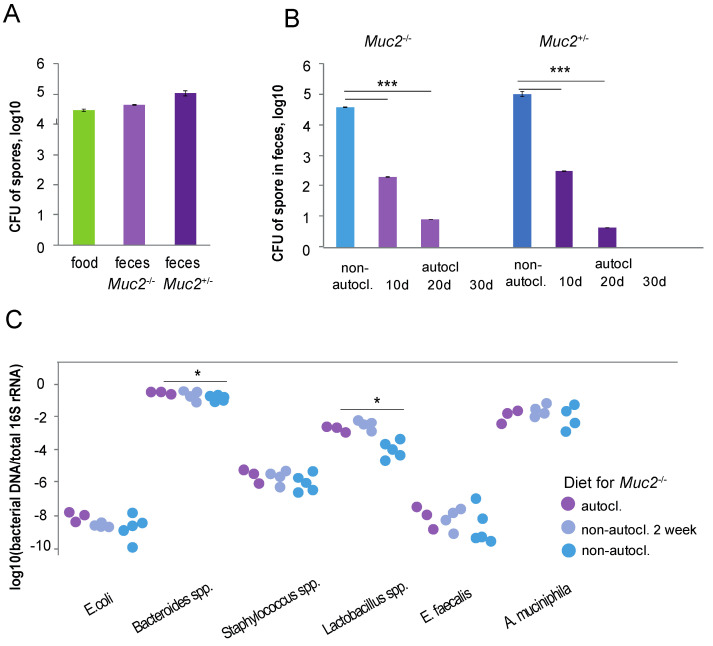
The *Bacillus* spp. spores in the diet and feces of mice. Amount of 16s rRNA gene of intestinal bacteria of mice fed on the autoclaved and non-autoclaved diets. (**A**) The *Bacillus* spp. spore in the diet and feces of *Muc2^−/−^* and *Muc2^+/−^* mice (*n* = 3), log10. (**B**) The number of spore in the feces of *Muc2^−/−^* and *Muc2^+/−^* mice on not autoclaved, 10 days autoclaved, 20 days autoclaved, and 30 days autoclaved food (*n* = 3 for each group). (**C**) Amount of 16s rRNA gene of intestinal bacteria (*Akkermansia muciniphila*, *Enterococcus faecalis*, *Lactobacillus murinus*, *Bacteroides* spp., *Staphylococcus* spp., *Escherichia coli*) of *Muc2^−/−^* mice fed on the autoclaved or non-autoclaved diet since birth, and 2 weeks feeding of the autoclaved diet of mice after eating the non-autoclaved diet since birth. *** *p* < 0.001, Fisher’s exact test, * *p* < 0.05, Mann–Whitney *U*-test.

**Figure 3 animals-12-02399-f003:**
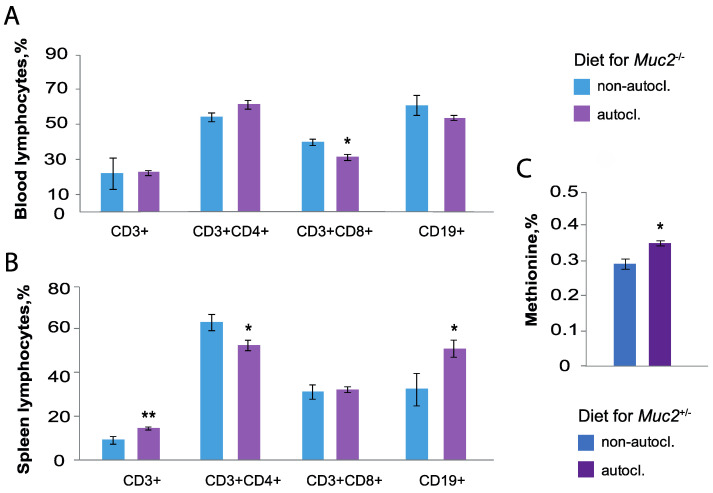
Percent of immune cells of *Muc2^−/−^* mice fed on the autoclaved and non-autoclaved diets. (**A**) The percent of immune cells in the blood of mice fed on the autoclaved (*Muc2^−/−^ n* = 5) and non-autoclaved diets (*Muc2^−/−^ n* = 4). (**B**) The percent of immune cells in the spleen of mice fed on the autoclaved (*Muc2^−/−^ n* = 5) and non-autoclaved diets (*Muc2^−/−^ n* = 5). (**C**) The methionine level in thigh muscle of mice (*Muc2^+/−^* on the autoclaved diet *n* = 5, *Muc2^+/−^* on the non-autoclaved diet *n* = 5). * *p* < 0.05, ** *p* < 0.01 Mann–Whitney *U*-test.

**Table 1 animals-12-02399-t001:** Amylase, protease, and lipase activities of bacterial strains isolated from the diet and feces.

Source	Bacterial Strains		Enzymatic Activity, Zone Diameter		
	No	Description	Amylase, mm	Protease, mm	Lipase
Diet	1	*Bacillus subtilis* strain qx-4, *Endophytic bacterium* MD3	37.7 ± 1.5	28.7 ± 0.3	+
	2	*Bacillus thuringiensis* strain GZDF1, *Bacterium* strain MIS_YL_J55	33.0 ± 1.0	20.3 ± 0.3	−
	3	*Lysinibacillus* spp. FWQSR5, *Lysinibacillus macroides* strain Z010	26.7 ± 0.9	−	−
	4	*Bacillus cereus* strain S43, *Bacillus thuringiensis* strain BT62 chromosome, *Bacillus paramycoides* strain	23.0 ± 1.5	28.0 ± 0.7	−
Feces	1	*Bacillus amyloliquefaciens* strain QT-162, *Bacillus velezensis* strain 2645	28.3 ± 0.9	28.0 ± 0.6	−
	2	*Bacillus thuringiensis* strain GZDF1, *Bacillus* sp. *(in: Bacteria)* strain 201705CJKOP-59	27.3 ± 1.5	22.3 ± 2.1	−
	3	*Bacillus subtilis* subsp. *stercoris* strain EGI137, *Bacillus subtilis* strain a22	24.0 ± 0.6	19.0 ± 0.6	+++
	4	*Bacillus cereus* strain YB1806, *Bacillus parathracis, Bacillus thuringiensis* strain NO.8	30.0 ± 0.0	22.3 ± 0.3	−
	5	*Bacillus cereus* strain D21, *Bacillus parathracis, Bacillus cereus* strain YB1806	−	21.0 ± 1.2	++++

**Table 2 animals-12-02399-t002:** Biochemical analysis of blood serum.

Metabolite	Group of Mice		Kolmogorov– Smirnov Test, *p*	Mann–Whitney *U* Test	
	Non-Autoclaved Diet	Autoclaved Diet		Z	*p*-Value
Createnin, µmol/L	36.92 ± 7.18	42.99 ± 11.14	*p* > 0.10	−0.31	0.75
Total protein, g/L	42.69 ± 1.09	41.81 ± 0.47	*p* > 0.10	0.73	0.46
LDL, µmol/L	2.57 ± 0.52	3.28 ± 0.82	*p* > 0.10	−0.31	0.75
Glucose, µmol/L	9.83 ± 0.29	8.84 ± 0.60	*p* > 0.10	1.15	0.25

## Data Availability

The data presented in this study are available on request from thecorresponding author.

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
