# Peer review of "Autoclaved Diet with Inactivated Spores of Bacillus spp. Decreased Reproductive Performance of Muc2−/− and Muc2+/− Mice"

_animals, 2022, doi:10.3390/ani12182399_

Round 1

Reviewer 1 Report

The authors examined an interesting question about the potential deleterious effects of raising mice in sterile environments. As barrier facilities work to keep pathogen and microbial loads in check, many sterilization practices mean that animals are not introduced to environmental microbes that can be beneficial for the animal’s microbiome and thus their health and wellness. The authors examine the effect of the sterilized feed on the reproductive efficiencies of female MUC2 mice.

Major concerns

1. The conclusion—Bacillus in non-autoclaved diet improves reproductive performance of mice—is not directly supported by the study design. There are two key test groups missing from this study. In addition to the autoclaved and non-autoclaved diet, an autoclaved diet + Bacillus spores as well as a non-autoclaved diet + heat inactivated Bacillus spores should be added to the study.

Since the temperature and pressures of autoclaving diets will not only inactivate Bacillus but also destroy vitamins and other nutrients, the authors need to unequivocally show that it’s the heat inactivation of Bacillus spores that increases the prolapse frequency and not other potential effects of autoclaving the diet such as nutrient degradation. The authors’ recommendation that Bacillus probiotics be given to mice in barrier facilities should have been tested (autoclaved diet + Bacillus) in the study rather than just suggested in conclusion.

Also, nutritional analysis of the diets is needed to show that presence/absence of Bacillus is the only change between the autoclaved and non-autoclaved diet.

2. 16S rRNA identification is a weak analysis for species identification and is insufficient for strain identification. Whole genome sequencing is preferred for stronger conclusions about strain identification. At the very least, additional sequence comparisons would strengthen the identification.  Section 3.4 states that each isolate had near 100% matches to multiple bacteria sometimes of different species and the authors arbitrarily strain level identification to each isolate.

3. The manuscript contains numerous English language and grammar issues that detract from the overall message, and in some cases, inhibit clear understanding.

Minor concerns

1. The introduction makes a case for the benefit of Bacillus based probiotics. However, the references cited generally refer to the benefit of Bacillus in plants and poultry, not in mice. It would make a stronger case to cite studies that show benefit of Bacillus to rodents specifically. Additionally, the references refer to the benefits of Bacillus subtilis, amyloliquefaciens, and licheniformis while more than half of the Bacillus found in the non-autoclaved diet and the feces of this study are from the Bacillus cereus group which are known to be pathogenic to animals. The references in the introduction and throughout should also address the advantages and disadvantages of the presence of Bacillus cereus group organisms.

2. Error bars not included on bar graphs. Include error bars in figures 1A and 1C.

3. Address why a different number of mice being analyzed in each aspect of the study. 5 mice were fed a non-autoclaved diet from birth while only 3 mice were fed an autoclaved diet from birth. Reproductive performance autoclaved (Muc2+/- n=74, Muc2-/- n=34) and non-autoclaved diet (Muc2+/- n=21, Muc2-/- n=10). In supplementary data 7, 8, 9, 10, and 11 mice were used for various studies.

4. Within the discussion it is stated that “It should be noted that the same species were found both in feces and in diet. This confirms that non-autoclaved diet is the source of Bacillus in mouse feces.” The data presented does not support that conclusion.

Per table 1: Amyloliquefaciens is found in the feces but not in the diet.

Diet

Feces

Bacillus subtilis strain qx-4

Bacillus subtilis subsp. stercoris strain EGI137

Bacillus thuringiensis strain GZDF1

Bacillus thuringiensis strain GZDF1

Lysinibacillus sp. FWQSR5

Bacillus amyloliquefaciens strain QT-162

Bacillus cereus strain S43

Bacillus cereus strain YB1806

Bacillus cereus strain D21

Additionally, lines 379 through 384 say that B. cereus was only found in the feces while all other B. cereus group isolates were actually B. thuringiensis. This statement is in direct opposition to the conclusion that all Bacillus present in the feces came from the diet. If B. thuringiensis was in the diet, authors need to address how B. cereus ended in the feces.

5. Figure 3 is labeled “Figure 2” above the figure.

Author Response

Dear editor,

Accompanying, please find our manuscript entitled: “Autoclaved diet with inactivated spores of Bacillus spp. decreased reproductive performance of Muc2-/- and Muc2+/- mice” by Morozova et al., which we would like to be secondly considered for publication as an Original article in Animals after adding new experiment, improving text according the reviewers comments.

We are grateful to the reviewers for analyzing our manuscript. Below we provide responses to all reviewer comments. The yellow color marks the text corrected in accordance with the comments of the reviewers. Green highlight text shows the edit of English languages.

Major concerns

  1. The conclusion—Bacillusin non-autoclaved diet improves reproductive performance of mice—is not directly supported by the study design. There are two key test groups missing from this study. In addition to the autoclaved and non-autoclaved diet, an autoclaved diet + Bacillus spores as well as a non-autoclaved diet + heat inactivated Bacillus spores should be added to the study.

    Since the temperature and pressures of autoclaving diets will not only inactivate Bacillus but also destroy vitamins and other nutrients, the authors need to unequivocally show that it’s the heat inactivation of Bacillus spores that increases the prolapse frequency and not other potential effects of autoclaving the diet such as nutrient degradation. The authors’ recommendation that Bacillus probiotics be given to mice in barrier facilities should have been tested (autoclaved diet + Bacillus) in the study rather than just suggested in conclusion.

    Also, nutritional analysis of the diets is needed to show that presence/absence of Bacillus is the only change between the autoclaved and non-autoclaved diet.

  • We agree with the reviewer. We are currently conducting further studies on the effects of Bacillus on behavior, immune performance, gut health, etc. We currently have data on Muc2-/- females fed autoclaved diet or autoclaved diet supplemented with spores of B. subtilis 5x109

In females on autoclaved diet, the first case of prolapse was registered at 2.5 months and by 6 months more than 80% of females had prolapse. In females supplemented with B. subtilis, no prolapse was noted by 8 months (n=10 per group). We also have Muc2+/- male mesenteric lymph node cell data. When B. subtilis is added to autoclaved diet, the number of cytotoxic and regulatory cells changes, which can affect the severity of inflammation if it is present. However, this is a large work that still needs a lot of time to complete and deserves a separate publication. In this article, we wanted to draw attention to the fact that food autoclaving can adversely affect the reproductive functions of laboratory animals. Therefore, we have removed the untested recommendations for the use of bacillus-based probiotics in autoclaved animals from this article. This is the last sentence in Simple Summary: "Addition of probiotic Bacillus can correct this effect on reproductive performance from an autoclaved diet" and in Conclusions: "It can be recommended for mice feed autoclaved diet and housed in barrier facilities to use biological additives based on bacteria of probiotic species of the genus Bacillus equivalent to the natural content in food- 103-104 CFU/g. Spores of B. subtilis or B. amyloliquefaciens can be the basis of such additives”.

Lines 21-22: We replaced in Simple Summary “We hypothesize that this is due the inactivation of spores of Bacillus spp. in the autoclaved diet” to: “We hypothesize that this is due to reducing the nutritional value of diet and the inactivation of spores of Bacillus spp. in the autoclaved diet."

Lines 651-652: In Conclusions we added: "We believe that further research on the addition of Bacillus probiotic cultures or their metabolites to autoclaved laboratory animal diet would be interesting and useful."

Of course, autoclaving degrades the nutritional value of the feed. Any heat treatment destroys vitamins. This negatively affects animal health. However, we have had experience in breeding mice using quality Sniff chow, which is made to be autoclaved. Additionally, the diet of mice was enriched with high-protein dog chow. This made it possible to slightly increase the reproductive capabilities of female mice, but the reproductive index was still lower than that of mice even on lower quality non-autoclaved food. The age of manifestation of prolapse was also earlier and enrichment with nutrients did not affect this parameter. From which we assume that the absence of Bacillus affects the manifestation of these signs, but this requires further verification. What we are already doing.

Lines 256-268: We analyzed the feed for amino acid content and chemical composition. Methods added to section 2. Materials and Methods (2.11. Analysis of the mineral and chemical composition of the diet). The results are described in Section 3.9.

Lines 496-508: Analysis of the mineral and chemical composition of the diet.

Lines 607-609: In discussion after the sentence “It is known that high levels of methionine can reduce autophagy and accelerate aging [71, 72]” added " The increase in methionine in mice fed on autoclaved diets does not depend on changes of the methionine level in the diet, since after autoclaving the level of methionine was even slightly lower in the diets"

Lines 643-650: Added to Conclusion: “It is known that autoclaving reduces the nutritional value of feed, in particular, destroys vitamins. However, our results did not reveal significant reductions in amino acids, crude protein, crude fat, etc. in feed after autoclaving. Only methionine, tryptophan and raw ash levels were reduced at trend level. It can be assumed that the reduced reproductive function of mice on autoclaved food may be affected by both factors: a relative decrease in the nutritional value of feed and inactivation of spores of beneficial Bacillus strains. The age of onset of prolapse is most likely significantly affected by the presence or absence of Bacillus spores.”

  1. 16S rRNA identification is a weak analysis for species identification and is insufficient for strain identification. Whole genome sequencing is preferred for stronger conclusions about strain identification. At the very least, additional sequence comparisons would strengthen the identification. Section 3.4 states that each isolate had near 100% matches to multiple bacteria sometimes of different species and the authors arbitrarily strain level identification to each isolate.identification is a weak analysis for species identification and is insufficient for strain identification. Whole genome sequencing is preferred for stronger conclusions about strain identification. At the very least, additional sequence comparisons would strengthen the identification. Section 3.4 states that each isolate had near 100% matches to multiple bacteria sometimes of different species and the authors arbitrarily strain level identification to each isolate.

-          Whole genome sequencing is preferred for determining bacterial strains, but is generally used to identify the most important bacteria. In this article, bacteria were cultured from feed and faeces, followed by identification by 16S rRNA sequencing. Since the conditions and raw materials for making the feed we use may differ in different batches, the bacteria can also be of different strains. There is no point in doing their whole genome sequencing with such an expensive method. We believe that the accuracy of 16S rRNA analysis is sufficient for the purposes of our article. Indeed, the read gene sequences could be attributed to different strains of bacteria, however, they all belonged to Bacillus spp. Absolute identification has no scientific value, since the bacterial load may be different in a different batch of feed. Our goal was to show that both food and faeces contain spores of Bacillus spp. and possible strains of bacteria that can inseminate the feed.

  1. The manuscript contains numerous English language and grammar issues that detract from the overall message, and in some cases, inhibit clear understanding.

-          We agree with the reviewer. We will use Language Editing Services.

Minor concerns

  1. The introduction makes a case for the benefit of Bacillus based probiotics. However, the references cited generally refer to the benefit of Bacillusin plants and poultry, not in mice. It would make a stronger case to cite studies that show benefit of Bacillusto rodents specifically. Additionally, the references refer to the benefits of Bacillus subtilisamyloliquefaciens, and licheniformis while more than half of the Bacillus found in the non-autoclaved diet and the feces of this study are from the Bacillus cereus group which are known to be pathogenic to animals. The references in the introduction and throughout should also address the advantages and disadvantages of the presence of Bacillus cereus group organisms.

  • We have added articles done in mice doi.org/10.3892/etm.2016.3686 doi: https://doi.org/10.1021/acs.jafc.1c03375.

Indeed, Bacillus-based probiotics were developed and used primarily in agriculture as a viable alternative to antibiotics. However, there are very few studies in laboratory animals. We believe that this is the weak point of current commercial Bacillus-based probiotics. In this regard, our work on mice may be of additional interest precisely in connection with the fact that it was performed on laboratory mice.

Lines 69-72: Added: Among the group of bacteria Bacillus cereus there are representatives that produce toxins and cause poisoning. However, other members of this group have a positive effect on health and are included in commercial probiotic preparations [13–15].

  1. Error bars not included on bar graphs. Include error bars in figures 1A and 1C.

It is corrected. For the fertility index, error bars cannot be shown, since this is a percentage of the entire sample.

  1. Address why a different number of mice being analyzed in each aspect of the study. 5 mice were fed a non-autoclaved diet from birth while only 3 mice were fed an autoclaved diet from birth. Reproductive performance autoclaved (Muc2+/- n=74, Muc2-/- n=34) and non-autoclaved diet (Muc2+/- n=21, Muc2-/- n=10). In supplementary data 7, 8, 9, 10, and 11 mice were used for various studies.

This study was not carried out within the framework of our main scientific activities. This work is carred out as a consequence of our observation that when changing diet from autoclaved to non-autoclaved, the reproductive index is increased and age of rectal prolapse manifistation is decreased. To calculate the reproductive index (Muc2+/- n=74, Muc2-/- n=34) on autoclavable diet and on non-autoclavable diet (Muc2+/- n=21, Muc2-/- n=10) data from breeding records for the same period of time were used. However, at present all of our animals are kept on non-autoclaved diet. To study the effect of autoclaved diet on the studied parameters of mice, we specially created a small colony of mice. We were guided by the principles of 3R regarding the use of the minimum required number of mice and the conduct of several experiments on the same animals, if the experimental conditions do not affect the results. Therefore, the number of animals in groups depended on the presence of female mice. We do not see the need to increase the samples, since the results obtained do not raise serious questions.

  1. 4. Within the discussion it is stated that “It should be noted that the same species were found both in feces and in diet. This confirms that non-autoclaved diet is the source of Bacillus in mouse feces.” The data presented does not support that conclusion.

Per table 1: Amyloliquefaciens is found in the feces but not in the diet.

Diet

Feces

Bacillus subtilis strain qx-4

Bacillus subtilis subsp. stercoris strain EGI137

Bacillus thuringiensis strain GZDF1

Bacillus thuringiensis strain GZDF1

Lysinibacillus sp. FWQSR5

Bacillus amyloliquefaciens strain QT-162

Bacillus cereus strain S43

Bacillus cereus strain YB1806

Bacillus cereus strain D21

Additionally, lines 379 through 384 say that B. cereus was only found in the feces while all other B. cereus group isolates were actually B. thuringiensis. This statement is in direct opposition to the conclusion that all Bacillus present in the feces came from the diet. If B. thuringiensis was in the diet, authors need to address how B. cereus ended in the feces.

Lines 586-592: Now the phrase “It should be noted that the same species were found both in feces and in diet. This confirms that non-autoclaved diet is the source of Bacillus in mouse feces “ is replaced with: “Sanger sequence has shown that feces and food contain the bacteria Bacillus spp. The microbiota composition of different aliquots of food and feces may differ. Also, since colonies were manually selected for sequencing by morphotype and microscopy, visually similar colonies might not be included in the analysis. There have been shown that non-autoclaved bedding is free of Bacillus spp. bacteria. It is also known and our results confirm that Bacillus spp. are not residents of the digestive tract, but enter it from the external environment. We believe that the sequencing results show the possible diversity of Bacillus spp. in diet.

  1. Figure 3 is labeled “Figure 2” above the figure.

It is corrected.

Reviewer 2 Report

animals-1856579-peer-review-v1

This is interesting research project with some clear conclusions and application for the caring of experimental animals.

I would like to recommend this work to be accepted for publication, however, after appropriate corrections, adjustments and clarifications.

Specific comments:

Ln41: Please, check if the last keyword is whitened correct.

Ln60: Please, maybe you can consider to replace "probiotic" with "beneficial".

Ln60: Maybe authors will consider to start story about Bacillus as new paragraph. 

Ln73: consider to start new paragraph with story about mucin.

Ln89: "To obtain Muc2+/+ and Muc2+/+ mice " is this correct? Check +/+?

Ln157-164: Maybe an appropriate reference can be added for the presented methods?

Ln164: CaCl2, 2 needs to be in index position.

Ln173: Please, add USA after CA. For all suppliers of material and equipment, please, provide address according to the standards: Name of the Company, City, State (in case of federal countries) and name of the country. Please, try to use address of the headquarter and not for the distributing companies.

Ln194: CO2, 2 need to be in index position. Please, check entier manuscript for similar adjustments.

Ln197: Add USA after NY., similar adjustments for the all paper, see Ln202 as well.

Please, be sure that information published previously under paper 29, is not entering with conflict in the present paper. Presume that this is same type of experiment preformed in different occasion and associated with different set of experiments. In case of conflict, please, remove appropriate information from this paper, including material and methods and results and discussion and refer to previously published paper (29).

Section 3.4. Are these Bacillus species were recorded in the animal feed?

Ln407: Please, correct the spelling of Bacillus

Ln409: spelling of E. faecalis.

Please, for all bacterial names if they are mentioned for first time, a full name needs to be provided. In second occasion, abbreviated according international standards need to be applied.

Ln409: spp. do not need to be in italics.

Table 1. Please, revise use of italics in the table 1. Strain identifications do not need to be in italics, word "strain" do not need to be in italics.

Ln460: please, remove one of the ".".

Ln538: Correct to E. coli

Ln568: Correct to "... more [70]."

Ln570-573: Please, abbreviate names, most of them are already introduced in the text earlier.

In references, please, pay additional attention to use of italics, abbreviation of journal names, use of DOI or doi, etc

Author Response

Dear editor,

Accompanying, please find our manuscript entitled: “Autoclaved diet with inactivated spores of Bacillus spp. decreased reproductive performance of Muc2-/- and Muc2+/- mice” by Morozova et al., which we would like to be secondly considered for publication as an Original article in Animals after adding new experiment, improving text according the reviewers comments.

We are grateful to the reviewers for analyzing our manuscript. Below we provide responses to all reviewer comments. The yellow color marks the text corrected in accordance with the comments of the reviewers. Green highlight text shows the edit of English languages.

Specific comments:

Line 41: Please, check if the last keyword is whitened correct.

We changed Muc2-/ mice- to Muc2-/- mice

Line 60: Please, maybe you can consider to replace "probiotic" with "beneficial".

We replaced "probiotic" with "beneficial".

Line 22-23: “Addition of beneficial Bacillus can correct this effect on reproductive performance from an autoclaved diet.”

Line 60: Maybe authors will consider to start story about Bacillus as new paragraph.  

It is corrected.

Line 73: consider to start new paragraph with story about mucin.

It is corrected.

Line 89: "To obtain Muc2+/+ and Muc2+/+ mice " is this correct? Check +/+?

The phrase was corrected: To obtain Muc2+/- and Muc2-/-mice”.

Lines 157-164: Maybe an appropriate reference can be added for the presented methods? 

We have provided references to methods if we were guided by specific previously published methods. In the event that the methods used are widely used or are with our modifications and were not previously published by us in international journals, references are not presented.

Line 164: CaCl2, 2 needs to be in index position.

It is corrected.

Line 173: Please, add USA after CA. For all suppliers of material and equipment, please, provide address according to the standards: Name of the Company, City, State (in case of federal countries) and name of the country. Please, try to use address of the headquarter and not for the distributing companies.

It is corrected: (Optimice, Animal Care Systems, Colorado, USA); (R-22, BioPro, Novosibirsk, Russia), (Anavidin, Irkutsk, Russia), (Getinge Sterilization AB, Gettinge, Sweden), (Merck, Darmstadt, Germany), (Olympus Corp., Tokyo, Japan), (Milli-Q type I, Merck Millipore, Darmstadt, Germany), (Bio-Rad, California, USA),  (Applied Biosystems, Massachusetts, USA), (Applied Biosystems, Massachusetts, USA), (Lumex, Irkutsk,Russia), (SoyuzKhimProm, Novosibirsk, Russia), (catalog no. LAA21-1KT, Sigma Aldrich, Darmstadt, Germany), and etc.

Line 194: CO2, 2 need to be in index position. Please, check entier manuscript for similar adjustments. 

It is corrected.

Line 197: Add USA after NY., similar adjustments for the all paper, see Ln202 as well.

It is corrected.

Please, be sure that information published previously under paper 29, is not entering with conflict in the present paper. Presume that this is same type of experiment preformed in different occasion and associated with different set of experiments. In case of conflict, please, remove appropriate information from this paper, including material and methods and results and discussion and refer to previously published paper (29).

All the data presented in this work are new and have not been published before. The previously published work is a small publication of preparatory validation studies.

“published earlier”  was changed to “ more detailed study of diet has been published earlier”.

Section 3.4. Are these Bacillus species were recorded in the animal feed?

Yes, there are studies that have found representatives of Bacillus spp. in  farm animal feed. For example https://doi.org/10.3389/fmicb.2021.7831

Line 407: Please, correct the spelling of Bacillus

It is corrected.

Line 409: spelling of E. faecalis.

It is corrected.

Please, for all bacterial names if they are mentioned for first time, a full name needs to be provided. In second occasion, abbreviated according international standards need to be applied. 

It is corrected.

Line 409: spp. do not need to be in italics. 

It is corrected.

Table 1. Please, revise use of italics in the table 1. Strain identifications do not need to be in italics, word "strain" do not need to be in italics.

It is corrected.

Line 460: please, remove one of the ".".

It is corrected.

Line 538: Correct to E. Coli

It is corrected.

Line568: Correct to "... more [70]." 

It is corrected.

Ln570-573: Please, abbreviate names, most of them are already introduced in the text earlier. 

It is corrected.

In references, please, pay additional attention to use of italics, abbreviation of journal names, use of DOI or doi, etc

The references were created using the Mendeley program and indeed, not all of them were correct. Now all references were corrected manually.

Round 2

Reviewer 1 Report

The authors have addressed my initial major concerns. While I would still prefer to see the additional work with supplementing Bacillus back into the diet, I can appreciate that the authors' are completing this work now and would like to save it for an additional manuscript.

The addition of the diet mineral and chemical analysis (Table S4) is helpful and further supports the authors discussion and conclusions about the effects of Bacillus spores.

The added reference Zhu, K. et al., Probiotic Bacillus cereus strains, a potential risk for public health in China, Front. 2016, Microbiol., 7, doi: 695 10.3389/fmicb.2016.00718. speaks to the dangers of B. cereus being used as a probiotic and does not necessarily support the benefits of B. cereus. In your future work, be sure to check for toxin production before using especially B. cereus as a mouse feed additive. 

Line 325 and 329 both contain the phrase "a more detailed study of the diet was published earlier [32]" This phrase seems misplaced in both cases and I'm not certain the reference is correct. Do you mean reference 33 instead of 32?

I acknowledge the effort to improve the English grammar. Another round of editing is required throughout the manuscript.

Author Response

We are grateful to the reviewers for analyzing our manuscript. Below we provide responses to comments. The yellow color marks the text shows the edit of English languages.

The added reference Zhu, K. et al., Probiotic Bacillus cereus strains, a potential risk for public health in China, Front. 2016, Microbiol., 7, doi: 695 10.3389/fmicb.2016.00718. speaks to the dangers of B. cereus being used as a probiotic and does not necessarily support the benefits of B. cereus. In your future work, be sure to check for toxin production before using especially B. cereus as a mouse feed additive. 

Study of the effects of Bacillus on the health of laboratory mice is ongoing, but the data we have already obtained indicate the need for more detailed and thorough research and discussion of this topic than just additional studies for this article. We also agree with the reviewer that B. cereus may be dangerous to animals. Therefore, we believe that it is advisable to autoclave the feed for laboratory animals, and then enrich it with a specific probiotic culture. We are currently investigating one of the B. subtilis strains that is commercially available in our region. This strain has shown good results in farm animal studies. However, there are no data on the mechanism of action of these bacteria and the impact on the health and reproductive function of laboratory animals. For our research, we produce a bacterial culture ourselves. However, the commercial availability of the strain makes our work more relevant and has practical applications. We do not see the need to use B. cereus as a probiotic, since the choice of probiotic strains of Bacillus spp. is big.

Line 325 and 329 both contain the phrase "a more detailed study of the diet was published earlier [32]" This phrase seems misplaced in both cases and I'm not certain the reference is correct. Do you mean reference 33 instead of 32?

We thank the reviewer for their attention. Indeed, the order of some references was broken in the bibliography after manual proofreading. Now we have corrected it. The phrase "a more detailed study of the diet was published earlier [32]" was added in agreement with second reviewer recommendations.

I acknowledge the effort to improve the English grammar. Another round of editing is required throughout the manuscript.

We did additional work to improve English by contacting a native English speaker.